# A Method for Broccoli Seedling Recognition in Natural Environment Based on Binocular Stereo Vision and Gaussian Mixture Model

**DOI:** 10.3390/s19051132

**Published:** 2019-03-06

**Authors:** Luzhen Ge, Zhilun Yang, Zhe Sun, Gan Zhang, Ming Zhang, Kaifei Zhang, Chunlong Zhang, Yuzhi Tan, Wei Li

**Affiliations:** College of Engineering, China Agricultural University, Qinghua Rd.(E) No.17, Haidian District, Beijing 100083, China; luzhenge@cau.edu.cn (L.G.); yangzhilun@cau.edu.cn (Z.Y.); 13811604659@163.com (Z.S.); allanopel@sohu.com (G.Z.); JackyM@cau.edu.cn (M.Z.); zhangkaifeiyx@163.com (K.Z.); liww@cau.edu.cn (W.L.)

**Keywords:** broccoli seedling, binocular stereo vision, semi-global matching, Gaussian mixture model, 3D point cloud

## Abstract

Illumination in the natural environment is uncontrollable, and the field background is complex and changeable which all leads to the poor quality of broccoli seedling images. The colors of weeds and broccoli seedlings are close, especially under weedy conditions. The factors above have a large influence on the stability, velocity and accuracy of broccoli seedling recognition based on traditional 2D image processing technologies. The broccoli seedlings are higher than the soil background and weeds in height due to the growth advantage of transplanted crops. A method of broccoli seedling recognition in natural environments based on Binocular Stereo Vision and a Gaussian Mixture Model is proposed in this paper. Firstly, binocular images of broccoli seedlings were obtained by an integrated, portable and low-cost binocular camera. Then left and right images were rectified, and a disparity map of the rectified images was obtained by the Semi-Global Matching (SGM) algorithm. The original 3D dense point cloud was reconstructed using the disparity map and left camera internal parameters. To reduce the operation time, a non-uniform grid sample method was used for the sparse point cloud. After that, the Gaussian Mixture Model (GMM) cluster was exploited and the broccoli seedling points were recognized from the sparse point cloud. An outlier filtering algorithm based on k-nearest neighbors (KNN) was applied to remove the discrete points along with the recognized broccoli seedling points. Finally, an ideal point cloud of broccoli seedlings can be obtained, and the broccoli seedlings recognized. The experimental results show that the Semi-Global Matching (SGM) algorithm can meet the matching requirements of broccoli images in the natural environment, and the average operation time of SGM is 138 ms. The SGM algorithm is superior to the Sum of Absolute Differences (SAD) algorithm and Sum of Squared Differences (SSD) algorithms. The recognition results of Gaussian Mixture Model (GMM) outperforms K-means and Fuzzy c-means with the average running time of 51 ms. To process a pair of images with the resolution of 640×480, the total running time of the proposed method is 578 ms, and the correct recognition rate is 97.98% of 247 pairs of images. The average value of sensitivity is 85.91%. The average percentage of the theoretical envelope box volume to the measured envelope box volume is 95.66%. The method can provide a low-cost, real-time and high-accuracy solution for crop recognition in natural environment.

## 1. Introduction

Broccoli is rich in nutrients and has a wide area of planting in China. However, in the field environment, the weeding and pesticide spraying of broccoli are still mainly manual. Therefore, it is urgent to develop intelligent weeding robot and target spraying equipment in China. The recognition and location of broccoli seedlings based on machine vision plays a decisive role in the development of intelligent weeding robots and target spraying equipment.

However, the natural field environment is unstructured and the illumination conditions are uncontrollable. The background of the soil is complex and changeable. The colors of weeds and crop are close, especially under weedy conditions. Besides, traditional 2D image processing technology has inherent defects, so it is hard for traditional 2D image processing technologies to identify crops in natural fields accurately and stably. In recent years, with the development and application of high performance computers, many experts and scholars have begun to explore the application of 3D stereo information in agriculture. The acquisition methods or equipment of 3D stereo information of plant in field can be divided into binocular stereo vision [1,2], multi vision [3,4], RGB-D camera [5,6], structured light [7], multispectral 3D vision system [8,9], laser scanning [10,11], etc. As 3D stereo information contains both RGB and position information of crop, it can be widely used in the identification [12] and positioning [13,14] of plants, phenotypic parameter acquisition [15,16] and so on.

In terms of the application of stereo vision in broccoli seedling recognition, Li et al. [17] introduced a method to identify broccoli seedlings and green bean plants based on 3D imaging under weedy condition. Firstly, images of broccoli seedling and green bean plants were taken from field via a 3D time-of-flight camera and sparse noise points were filtered out by means of height threshold. Then both 2D and 3D features were extracted to recognize broccoli seedling and green bean plants, but artificial threshold was needed throughout the whole plants identification process. Andujar et al. [18] used depth cameras to get structural parameters of broccoli under laboratory environment conditions to assess the growth state and yield of broccoli.

The 3D point cloud contains not only the RGB information of the plants but also their spatial position information, so in addition to broccoli seedlings, stereo vision technologies can be also applied to identification and localization of other crops in the field, in plant factorys or in the greenhouse. Avendano et al. [19] collected videos of coffee branches with a portable image acquisition device. Structure from Motion (SFM) and Patch Multi-View Stereo (PMVS) were developed to get the 3D point cloud of coffee branches, and Support Vector Machine (SVM) was applied to distinguish the six nutrient structures of coffee branches. Nguyen et al. [20] used a RGB-D camera to recognize and locate apples in apple orchard. Firstly, distance and color filters were exploited to remove points of leaves, branches, and trunks from original point cloud. Then Euclidean clustering algorithm was utilized in identifying apple points. Punica granatum was recognized and located by use of stereo vision [21]. Wang et al. [22] proposed a method of litchi fruit positioning in natural environment based on binocular stereo vision. Wavelet transform algorithm was used to unify the illumination of image. Then K-means algorithm was used to segment the Litchi fruit, and Normalized Cross-Correlation (NCC) algorithm was developed for stereo matching of Litchi fruit. 3D positioning information of Litchi fruit can be obtained by 3D reconstruction finally. Meanwhile, binocular stereo vision can also be used to identify and locate sweet pepper stems [23] and distinguish crops and weeds in the field [24]. Vazquez-Arellano et al. [25] used a time-of-flight camera for 3D reconstruction and positioning of maize in field. Experimental results show that the average error and standard deviation of the positioning of maize are 3.4 cm and ±1.3 cm, respectively. But the method proposed in this paper has a heavy computation. Mehta et al. [26] made use of multi-vision to recognize and locate fruit in 3D point cloud space. 

In recent years, more and more experts and scholars began to explore mounting a stereo vision imaging device on picking robots [27], automatic guided vehicles [28,29], unmanned aerial vehicles [30], and industrial manipulators [31], to acquire 3D models of plants more flexibly and conveniently. Then, these 3D models can be applied for plant organs extraction [32], 3D phenotypic parameters acquisition [33,34], crops monitoring [35], biomass assessment [30], pest detection [8,9], crop yield prediction [36], and crop growth database establishment [37]. All the works above were finally applied to guide agricultural production.

In terms of plants phenotypic parameters acquisition, stereo vision is widely used, because its advantages of non-destructive, non-contact and high precision. Hui et al. [38] proposed a method of plants phenotypic parameters acquisition and plant monitoring based on multi vision under laboratory conditions. Firstly, Multi-View Stereo and Structure from Motion (MVS-SFM) algorithm and VisualSFM software were used to obtain 3D point cloud of plant. Then, plant phenotypic parameters were calculated in 3D point cloud space. Hausdorff distance was calculated between phenotypic parameters and laser scanning. The results showed that the method introduced in this paper had a high precision. Li et al. [39] built a portable, low-cost binocular stereo vision system integrated by a network camera and a 3D time-of-flight (ToF) camera which can be applied for phenotypic parameters extraction of maize under laboratory conditions. An et al. [40] developed a system composed with 18 cameras to obtain images of plants in the greenhouse and the Agisoft PhotoScan software was used for 3D point cloud reconstruction. Then, the crop phenotypic parameters were acquired which were finally exploited for crop monitoring, but the system applied in this article is high cost and the method proposed in this paper has a heavy computation burden. Bao et al. [41,42] developed an imaging acquisition system composed of six pairs of binocular cameras for phenotypic parameter acquisition of sorghum crops in the field. A Semi-Global Matching (SGM) algorithm was used for binocular stereo matching in this paper. A hyperspectral pushbroom sensor unit was used for hyperspectral images acquisition of crop, and a perceptron laser triangulation scanner was applied for crop 3D modelling. Then, hyperspectral 3D plant models were obtained by fusing the spectral information and spatial information, and the models were used for crop phenotypic parameters acquisition, crop lesion identification, and crop tissue classification [43], finally. Golbach et al. [44] developed a system composed of 10 cameras for seedling phenotypic parameter measurement and a shape-from-silhouette method was exploited for 3D point cloud reconstruction. 

Santos et al. [45] used a handheld camera for crop image acquisition. Then a Multi-View Stereo and Structure from Motion (MVS-SFM) algorithm was developed to get crop 3D models and a spectral clustering algorithm was exploited for single blade identification. Finally, phenotypic parameters of single blades can be measured in 3D point cloud space. An optical sensor Artec Spider 3D scanner and 3D-Bunch-Tool software were exploited for obtaining phenotypic parameters of grapefruit under laboratory conditions [46]. Moriondo et al. [47] developed a method for phenotypic parameter acquisition of olive leaves based on stereo vision technologies under laboratory conditions. Agisoft PhotoScan software and the Structure from Motion (SFM) algorithm were proposed for olive tree 3D point cloud reconstruction, and a Random Forest algorithm was used to segment olive leaves points from olive 3D point cloud. Then a label connected components algorithm (CCA) was sued to identify single olive leaf. Finally, leaf area, leaf inclination, and leaf azimuth of each olive can be obtained. Duan et al. [48] used stereo vision technologies for 3D point cloud reconstruction and phenotypic parameters acquisition of wheat crop. Chaivivatrakul et al. [49] used stereo vision technologies for 3D point cloud reconstruction and phenotypic parameters acquisition of maize under laboratory conditions. Rose et al. [50] introduced a method for tomato fruit phenotypic parameter measurement under laboratory conditions based on a laser scanner. Pix4DMapper software and Multi-View Stereo and Structure from Motion (MVS-SFM) algorithm were made use for 3D point cloud reconstruction. After artificial denoising and segmentation of leaves and stems phenotypic parameters of leaves and stems, were calculated finally.

In summary, due to the extensive application of stereo vision in agriculture, a method of broccoli seedling recognition based on Binocular Stereo Vision and Gaussian Mixture Model was proposed in this paper: (1) Broccoli seedling images were acquired by a portable integrated binocular camera in the field under natural environment conditions; (2) the Matlab calibration toolbox was used for binocular camera calibration to obtain internal and external parameters of the binocular camera; (3) Epipolar rectification; (4) the Semi-Global Matching (SGM) algorithm was exploited to get disparity maps; (5) 3D point cloud reconstruction; (6) invalid point removal; (7) 3D point cloud down-sampling by using a non-uniform grid sample method; (8) broccoli seedling points were recognized by Gaussian Mixture Model Cluster; (9) broccoli seedling points were denoised by means of the k-nearest neighbors (KNN) algorithm. The aim of the method proposed in this paper is to solve the problem of broccoli seedling recognition in the field under natural environment conditions including different exposures, different weed conditions and different camera heights.

## 2. Materials and Methods

### 2.1. Image Acquisition and Experiment Platform

The broccoli seedlings were bred on March 25, 2018 and transplanted on April 28, 2018. The broccoli seedling images were acquired from 10:00–12:00 pm, on May 23, 2018, ate the Beijing International Urban Agricultural Science and Technology Park (116°47′57″E, 39°52′7″N). The image acquisition device is an integrated binocular camera (VR (Virtual Reality) Camera, BOBOVR, Shenzhen, China) with a resolution of 1280×480, the frame frequency of 30 fps, CMOS, Fov 120°, the baseline length of 60 mm, the working distance of 500–2000 mm, USB 2.0, 600 RMB. The working platform is OMEN by HP Desktop PC 880-p1xx, 8GB RAM, Inter Core i7-8700 @ 3.20 GHz, Windows 10, 64 bit system (DirectX 12). The software used was MATLAB R2016b (Math Works Corporation, Nattick, MA, USA), and Adobe Photoshop CS6 (64 bit, Adobe, San Jose, CA, USA). A checkerboard calibration board with a square size of 30 mm × 30 mm was used for camera calibration. The broccoli seedling images and binocular camera are shown in Figure 1.

### 2.2. Methods

#### 2.2.1. Binocular Camera Calibration

MATLAB Stereo Camera Calibrator APP [51] is exploited for binocular camera calibration and the intrinsic parameter matrix, distortion coefficient, essential matrix, fundamental matrix, rotation matrix and translation matrix of binocular camera are obtained, which can be used for binocular stereo rectification and broccoli seedling point cloud reconstruction. Checkerboard calibration board images are shown in Figure 2.

#### 2.2.2. Stereo Rectification

The purpose of stereo rectification is to eliminate the radial distortion and tangential distortion of the binocular images, and make the left and right images satisfy the epipolar constraint. It means that the same point of the same object is on the same horizontal line in the in the rectified images. So that the disparity searching range is changed from 2D planar search to 1D linear search. Bouguet’s stereo rectification method [52] was adopted.

#### 2.2.3. Semi-Global Matching Algorithm and 3D Recognition

The Semi-Global Matching (SGM) algorithm was used for stereo matching which was firstly proposed by Hirschmuller [53] in 2005. The disparity map was obtained after stereo matching. The disparity map and intrinsic parameter matrix of left camera were used for 3D recognition, and the original broccoli seedling point cloud can be obtained finally.

#### 2.2.4. Invalid Points Removal and Down-Sampling

There was a large amount of invalid points in the original point cloud which would be removed after Invalid points removal. Invalid points removal can reduce the point number of broccoli seedling point cloud, and the point cloud would be transformed from ordered point cloud into a disordered one. Then Non-uniform box grid filter [54] was used for down-sampling to reduce the point number of broccoli seedling point cloud.

#### 2.2.5. Gaussian Mixture Model cluster

Gaussian mixture model [55] is a linear combination of multiple Gaussian distribution functions. Let $\varphi =\left\{\varphi_{n}\right\},\;n=1,2,\cdots ,N$, represent the broccoli seedling point cloud obtained by down-sampling, then the Gaussian mixture model can be expressed as:(1)$$p\left(\varphi \right)=\overset{K}{\underset{k=1}{\sum }}\pi_{k}Ν\left(\varphi |\mu_{k},\Sigma_{k}\right),$$

Let $\gamma \left(z_{nk}\right)$ represent the posterior probability of point $\varphi_{n}$, which belongs to the *k*th cluster. The probability that the kth class is not selected the probability that the kth class is not selected. Then, $\gamma \left(z_{nk}\right)$ can be obtained by Bayes’ theorem:(2)$$\gamma \left(z_{nk}\right)=\frac{\pi_{k}Ν\left(\varphi_{n}|\mu_{k},\Sigma_{k}\right)}{\sum_{j=1}^{K}\pi_{j}Ν\left(\varphi_{n}|\mu_{j},\Sigma_{j}\right)},$$

Theoretically, $\mu_{k}$, $\Sigma_{k}$, $\pi_{k}$ can be obtained by using $\gamma \left(z_{nk}\right)$:(3)$$\mu_{k}=\frac{1}{\sum_{n=1}^{N}\gamma \left(z_{nk}\right)}\overset{N}{\underset{n=1}{\sum }}\gamma \left(z_{nk}\right)\varphi_{n},$$

(4)$$\Sigma_{k}=\frac{1}{\sum_{n=1}^{N}\gamma \left(z_{nk}\right)}\overset{N}{\underset{n=1}{\sum }}\gamma \left(z_{nk}\right)(\varphi_{n}-\mu_{k}){\left(\varphi_{n}-\mu_{k}\right)}^{T},$$

(5)$$\pi_{k}=\frac{\sum_{n=1}^{N}\gamma \left(z_{nk}\right)}{N},$$

The steps of Expectation-Maximization algorithm are as follows:
(1)Let K be the number of the cluster of the broccoli seedling point cloud, and set initial values of $\pi_{k}$, $\mu_{k}$, $\Sigma_{k}$ separately.(2)Calculate the posterior probability $\gamma \left(z_{nk}\right)$ by using Equation (2) according to the current $\pi_{k}$, $\mu_{k}$, $\Sigma_{k}$.(3)Calculate the new $\pi_{k}{}^{new}$, $\mu_{k}{}^{new}$, $\Sigma_{k}{}^{new}$ by using the Equations (3–5).(4)Calculate the logarithmic likelihood function of Equation (1).
(6)$$lnp\left(\varphi \right)=\overset{N}{\underset{n=1}{\sum }}ln[\overset{K}{\underset{k=1}{\sum }}\pi_{k}Ν\left(\varphi |\mu_{k},\Sigma_{k}\right)],$$(5)Check whether the parameters $\pi_{k}$, $\mu_{k}$, $\Sigma_{k}$ are convergent or the function (6) is convergent, if not return to (2).(6)If converge, calculate posterior probability $\gamma \left(z_{nk}\right)$ of each point of broccoli seedling point cloud separately, and then categorize the point to the cluster, where $\gamma \left(z_{nk}\right)$ has the maximum value.

#### 2.2.6. Outlier Filtering by K-Nearest Neighbors (KNN) Algorithm 

Broccoli seedling points would be recognized by using the Gaussian Mixture Model cluster. But there were still some outliers in the recognized broccoli seedling points, so the K-Nearest Neighbors (KNN) algorithm [56] would be exploited for outlier filtering, then an ideal broccoli seedling point cloud would be acquired finally.

## 3. Results

### 3.1. Stereo Rectification Analysis

Figure 3 shows that the distortion of original RGB images was eliminated in rectified images. The broccoli seedling occupies less area in rectified images because of the interpolation operation and image cutting in the stereo rectification process. After stereo rectification, the resolution of the image changes from 640 × 480 to 791 × 547.

### 3.2. Stereo Matching Results Analysis

As shown in Figure 4a–c, disparity maps can be obtained by the SGM algorithms. Figure 4a,c shows that the broccoli seedling regions were matched smoothly, and the marginal parts were also preserved completely. In Figure 4b, due to the camera overexposure, there are some mismatched areas in the upper and left blades of the broccoli seedlings, but the boundary between broccoli seedlings and background was matched clearly, and for this reason, these mismatched areas did not affect the next reconstruction and identification of broccoli seedling.

As shown in Figure 4d–i, disparity maps obtained by the SSD algorithm [57] are superior to disparity maps obtained by SAD algorithm [57], but their quality are both inferior to disparity maps obtained by SGM algorithm, because of the un-smoothness, ambiguous boundaries between broccoli seedling and background, and large noise areas of background shown as the orange, red and yellow areas. The matching window size, maximum disparity and matching time of SAD, SSD, SGM are shown in Table 1.

As shown in Table 1, as far as SAD algorithm and SSD algorithm are concerned, to obtain an ideal disparity map, a large matching window is required, so the matching window size of 55×55 pixel was selected. As can be seen in Figure 5, when the matching window size is 55×55 pixel and maximum disparity is 130 pixel, the best matching disparities for the SAD algorithm at point (400,400) pixel, are 89 pixel, 75 pixel, 45 pixel and for SSD algorithm are 90 pixel, 75 pixel, 45 pixel respectively. Therefore, in order to obtain ideal disparity maps of images with different shooting heights, a larger maximum disparity value should be selected. However, when the matching window size is certain, the operation time of SAD algorithm and SSD algorithm will increase with the increase of the maximum disparity. As shown in Table 1, when the matching window size is 55×55 pixel and the maximum disparities are 130 pixel, 120 pixel and 110 pixel respectively for images in Figure 1, the average operation time of SAD algorithm is 1475 ms and of SSD algorithm is 1498 ms. 

As far as the SGM algorithm is concerned, when the matching window size is 15 × 15 pixel, the maximum disparity is 128 pixel, the proposed SGM algorithm can satisfy different weed conditions, different shooting heights, and different exposure intensities, and ideal disparity maps can be obtained in real-time. The average operation time of SGM algorithm is 138ms which only accounts for 9.36% of the SAD algorithm, and 9.21% of SSD algorithm.

### 3.3. Reconstruction, Invalid Points Removal and Down-Sampling Results Analysis

As shown in Figure 6a–c, broccoli seedlings, weeds, pipelines, and soil were reconstructed successfully, and the height advantage of broccoli seedling is highlighted. But because of the matching error of SGM algorithm, there are still some outliers and invalid points in the original 3D point cloud.

The points of Inf were removed from original point cloud by an invalid point removal operation, at the same time, the ordered point cloud is transformed into a disorder one. As shown in Table 2, the number of points dropped from 430,000 to around 300,000 after invalid point removal. The point number of point cloud is significantly reduced, but maps in Figure 6a–c and Figure 6d–f look the same, that is because only valid points of 3D point cloud can be displayed. As can be seen in Figure 6g–i there is a distinct black boundary between broccoli seedlings and background. The reason is that the imaging principle of the camera is small hole imaging principle. When imaging, the broccoli seedling will occupy a larger region in the image plane, and a part of the background will be blocked by broccoli seedlings, because the broccoli seedlings are closer to the camera. After reconstructing, the size of broccoli seedlings become the actual size, and the occluded part becomes a black boundary region between the broccoli seedlings and background. This makes clustering and recognition of broccoli seedling points feasible and simple.

A non-uniform grid sample algorithm was adopted for point cloud down-sampling and the sparse point clouds were obtained, shown in Figure 6j–l. The point number of the sparse point cloud is 4,096, but all the characteristics of dense point cloud were completely preserved. The reduction of the point number will also reduce the computation time of the GMM algorithm simultaneously.

### 3.4. Broccoli Seedling Points Clustering and Recognition Results Analysis

As is shown in Figure 7a–c the GMM algorithm can recognize the broccoli seedling points from the point clouds in Figure 6j–l completely, when the component number of GMM is 10. Only broccoli seedlings can be recognized in Figure 7e by the K-means algorithm [58], and in Figure 7d,f there are a large amount of background points. 

As can be seen in Figure 7g,h, the broccoli seedling points can be recognized by the Fuzzy c-means algorithm [59], but in Figure 7h there are still a few background points. In Figure 7i, the recognition of broccoli seedling points by Fuzzy c-means failed. As can be seen in Figure 7f,i, the broccoli seedling points can’t be recognized by either the K-means algorithm or the Fuzzy c-means algorithm, when the shoot height is the highest, and the broccoli seedling occupies a smaller area in the image plane. In brief, the GMM algorithm is superior to the K-means algorithm and the Fuzzy c-means algorithm in terms of broccoli seeding recognition effect. There are still some outliers in broccoli seedling points recognized by GMM algorithm. Therefore, the KNN algorithm was used for outlier filtering. As shown in Figure 7j–l, all of the outliers were removed from broccoli seedling points and the detail of broccoli seedling was preserved fully, which can be seen in the red ellipse in Figure 7j,l.

As can be seen from Table 3, the average computation time of GMM algorithm is 51 ms, of K-mean algorithm is 8 ms, and of Fuzzy c-means is 173 ms which is 3.39 times longer than the GMM algorithm. The average computation time of K-mean algorithm is the shortest. However, the K-means algorithm is susceptible to outliers and has poor stability.

To further illustrate the stability of the GMM algorithm, 10 times clustering for each broccoli seedling were taken and the $\pi_{k}$ of the broccoli seedling component was obtained. A line diagram was drawn as shown in Figure 8. The standard deviation of the three sets of $\pi_{k}$ is 5.54 × 10^−4^, 6.24 × 10^−4^, 1.85 × 10^−3^ respectively, which shows the values of each set of $\pi_{k}$ have a small change. The three lines in Figure 8 are close to horizontal line, which shows the GMM algorithm has good stability.

### 3.5. Completeness of Broccoli Seedling Recognition

Furthermore, to illustrate the completeness of broccoli seedling recognition by the method proposed in this paper, considering that the application of machine vision of intelligent weeding robot and the target spraying equipment is crop identification and setting a protected area or a spraying area around the crop, sensitivity [60] was selected. Images were segmented manually, and the top view of broccoli seeding points obtained by the GMM algorithm are shown as Figure 9. The manual pixel area, the theoretical pixel area, and the intersection area are shown in Table 4. The average value of sensitivity is 85.91%, so the proposed method has a good completeness of broccoli seeding recognition.

### 3.6. Measured and Theoretical Envelope Box Volumes

To further illustrate the effectiveness and the universal adaptability of the algorithm proposed in this paper. A measured and theoretical envelope box of cabbage are shown in Figure 10, and the values of measured and theoretical envelope box volume are shown in Table 5. As can be seen in Table 4, the volumes of the measured and theoretical envelope box are very close, and the average percentage of theoretical volume to measured volume is 95.66%. The percentage of Figure 9b is 83.61%, because only the canopy height of a plant can be obtained by the proposed method, and the measured height of plant is the height of the plant to the ground. For all plants, the theoretical length and width are very close to the measured length and width, so the proposed method has a high canopy parameter acquisition precision.

## 4. Conclusions

(1)A method of broccoli seedling recognition was proposed in this paper, which is based on Binocular Stereo Vision and Gaussian Mixture Model clustering, under different weed conditions, different shooting heights, and different exposure intensities in a natural field. The method was proposed for the rapid identification of transplanted broccoli seedlings with growth advantage. The experimental results of 247 pairs of images proved that correct recognition rate of this method is 97.98%, and the average operation time to process a pair of original images with the resolution of 640×480 was 578 ms. The average value of sensitivity is 85.91%. For cabbage planta the average percentage of the theoretical envelope box volume to the measured envelope box volume is 95.66%.(2)The SGM algorithm was introduced for a pair of broccoli seedling images with the resolution of 791×547 after stereo rectification. The SGM algorithm was compared with the SAD algorithm and the SSD algorithm. The SGM algorithm can meet the matching requirements of all broccoli seedling images, when the matching window size was 15×15 pixel and the maximum disparity was 128 pixel. The operation time of SGM algorithm was 138 ms. The experimental results showed that SGM algorithm is superior to SAD algorithm and SSD algorithm.(3)The GMM cluster was adopted for recognizing broccoli seedling points rapidly and stably. The experimental results showed that the proposed GMM algorithm was better than the K-means algorithm and the fuzzy c-means algorithm on recognition effect and stability. The average calculation time of the GMM algorithm was only 51 ms which satisfied the real-time requirements. The KNN algorithm was used for outliers filtering of broccoli seedling points recognized by GMM cluster, and complete and pure broccoli seedling was recognized finally.

## Figures and Tables

**Figure 1 sensors-19-01132-f001:**
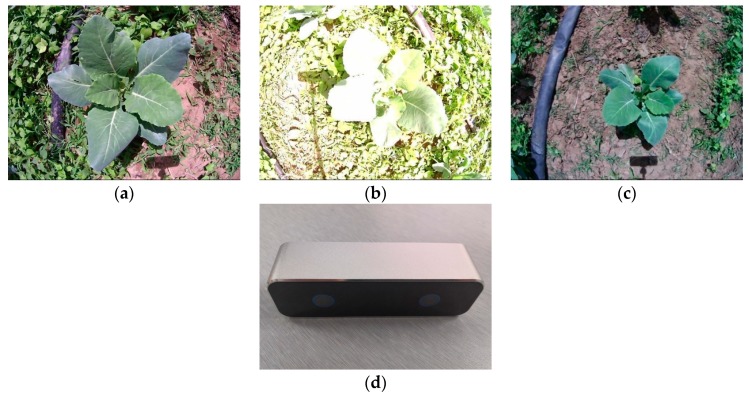
(**a**) represents the shooting height is the lowest, the amount of grass is larger, and the exposure intensity is moderate; (**b**) represents the shooting height is moderate, the amount of grass is the largest, and the exposure intensity is the largest; (**c**) represents the shooting height is the highest, the amount of grass is small, and the exposure intensity is normal; (**d**) represents the binocular camera (VR Camera).

**Figure 2 sensors-19-01132-f002:**
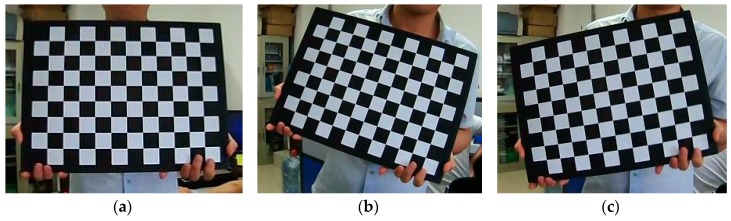
(**a**–**c**) represent the checkerboard calibration board images.

**Figure 3 sensors-19-01132-f003:**
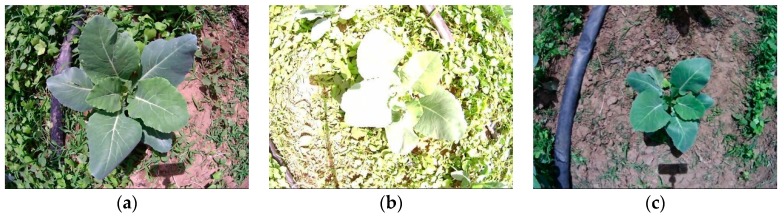
(**a**–**c**) represent the original left RGB image, (**d**–**f**) represent the corresponding rectified RGB images.

**Figure 4 sensors-19-01132-f004:**
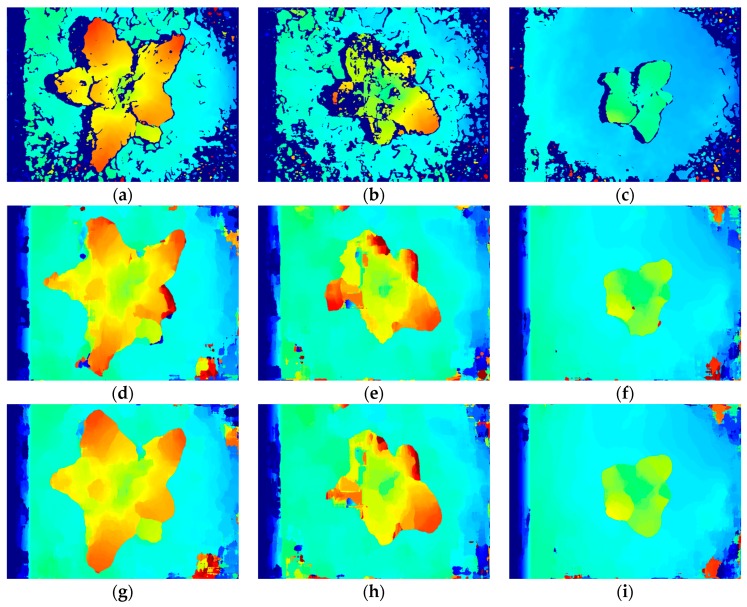
(**a**–**c**) represent disparity maps obtained by using the SGM algorithm, (**d**–**f**) represent disparity maps obtained by using the SAD algorithm, (**g**–**i**) represent disparity maps obtained using the SSD algorithm.

**Figure 5 sensors-19-01132-f005:**
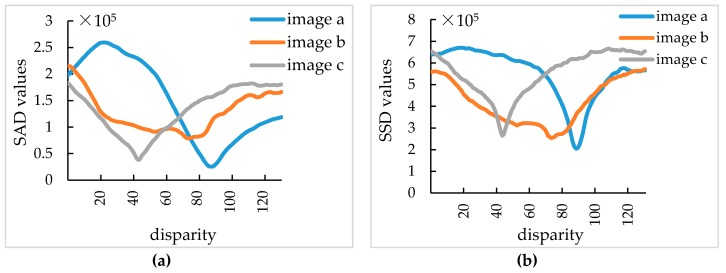
(**a**) represents SAD values when the stereo matching window size is 55×55 pixel, maximum disparity is 130 pixel at point (400,400) of images in Figure 3d–f; (**b**) represents SAD values when the stereo matching window size is 55×55 pixel, maximum disparity is 130 pixel at point (400,400) of images in Figure 3d–f.

**Figure 6 sensors-19-01132-f006:**
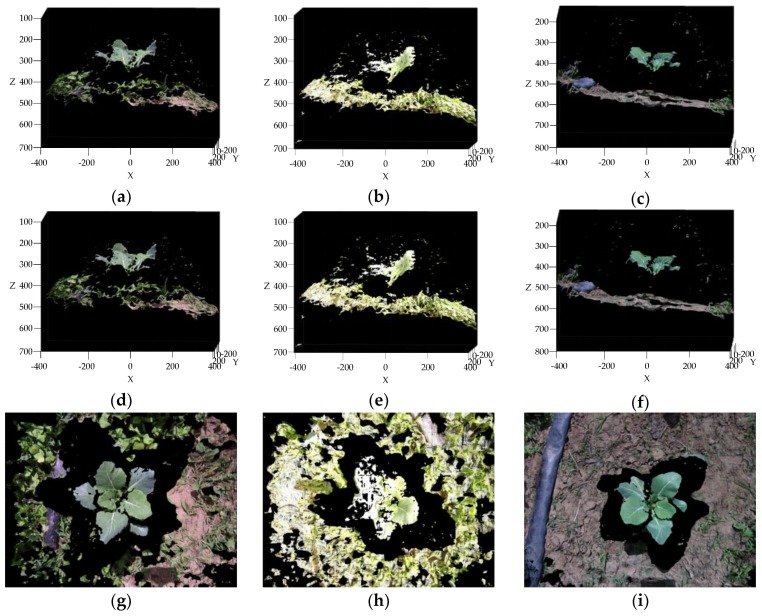
(**a**–**c**) represent the original 3D point cloud, (**d**–**f**) represent the 3D point cloud after invalid point removal of Figure 6a–c, (**g**–**i**) represent the top view of Figure 6d–f, (**j–l**) represent the sparse point cloud after down-sampling of Figure 6d–f.

**Figure 7 sensors-19-01132-f007:**
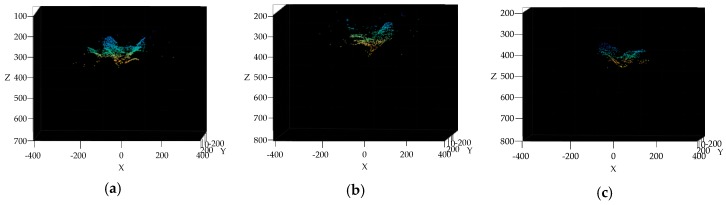
(**a**–**c**) represent broccoli seedling clustering results by the GMM algorithm, (**d**–**f**) represent broccoli seedling clustering results by the K-means algorithm, (**g**–**i**) represent broccoli seedling clustering results by the Fuzzy c-means algorithm, (**j**–**l**) represent outlier filtering results of Figure 7a–c by the KNN algorithm.

**Figure 8 sensors-19-01132-f008:**
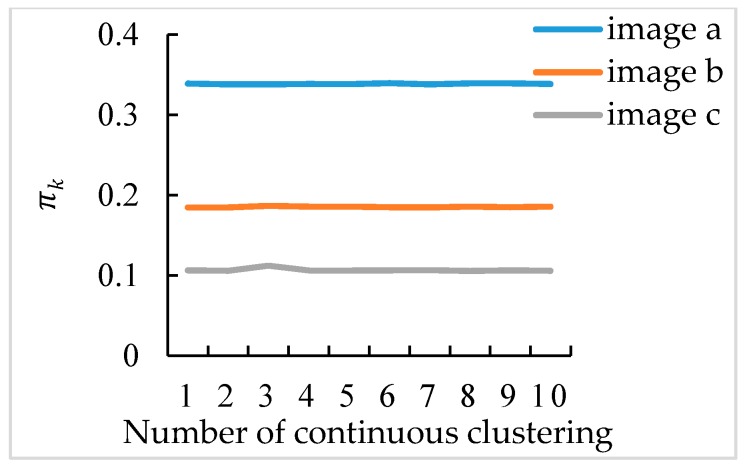
$\;\pi_{k}$ of broccoli seedling component.

**Figure 9 sensors-19-01132-f009:**
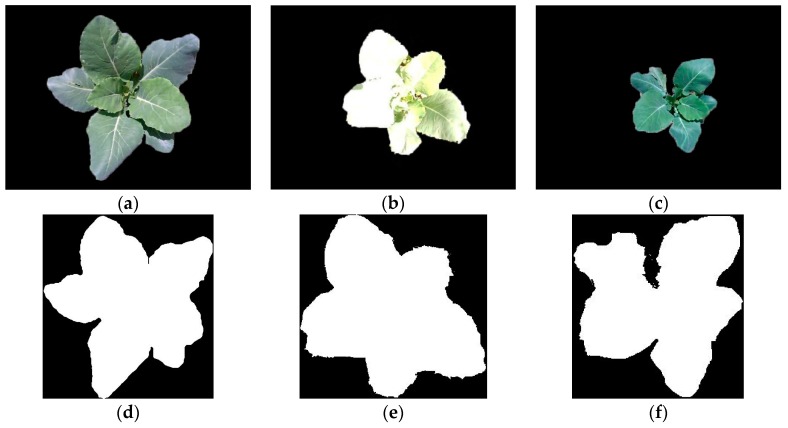
(**a**–**c**) Manually segmented images. (**d**–**f**) show cropped binary images of (**a**–**c**), (**g**–**i**) show top views of broccoli seeding points obtained by the GMM algorithm, (**j**–**l**) are binary images of (**g**–**i**).

**Figure 10 sensors-19-01132-f010:**
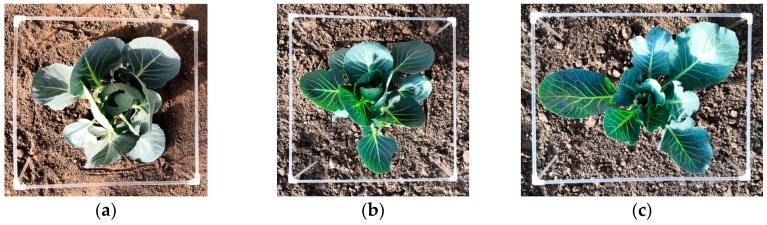
(**a**–**c**) represent envelope boxes for practical measurement, (**d**–**f**) represent envelope boxes acquired theoretically.

**Table 1 sensors-19-01132-t001:** Matching window size, maximum disparity and matching time of SAD, SSD, and SGM.

Image	SAD			SSD			SGM		
	Matching Window Size (Pixel)	Maximum Disparity (Pixel)	Matching Time (ms)	Matching Window Size (Pixel)	Maximum Disparity (Pixel)	Matching Time (ms)	Matching Window Size (Pixel)	Maximum Disparity (pixel)	Matching Time (ms)
a	55×55	130	1598	55×55	130	1610	15×15	128	142
b	55×55	120	1472	55×55	120	1502	15×15	128	135
c	55×55	110	1356	55×55	110	1383	15×15	128	138

**Table 2 sensors-19-01132-t002:** Point number of point cloud.

Image	Point Number of Original Point Cloud	Point Number of Point Cloud after Invalid Points Removal	Point Number of Sparse Point Cloud
a	432,677	296,053	4096
b	432,677	277,858	4096
c	432,677	340,974	4096

**Table 3 sensors-19-01132-t003:** Running time of three cluster.

Image	GMM (ms)	K-means (ms)	Fuzzy c-means (ms)
a	51	6	171
b	52	7	176
c	49	11	172

**Table 4 sensors-19-01132-t004:** Manual area and theoretical area.

Image	Area of Broccoli Seeding Obtained Manually (Pixel)	Area of Broccoli Seeding Obtained Theoretically (Pixel)	Intersection Area of Broccoli Seeding Obtained Manually and Theoretically (Pixel)	Sensitivity
a	1.08 × 10^5^	9.76 × 10^4^	8.48 × 10^4^	86.91%
b	7.00 × 10^4^	6.48 × 10^4^	5.74 × 10^4^	82.07%
c	3.79 × 10^4^	3.73 × 10^4^	3.36 × 10^4^	88.75%

**Table 5 sensors-19-01132-t005:** Measured and theoretical envelope box volumes.

Plants	Measured				Theoretical				(Theoretical Volume)/(Measured Volume)
	Length (mm)	Width (mm)	Height (mm)	Volume (mm^3^)	Length (mm)	Width (mm)	Height (mm)	Volume (mm^3^)	
a	263.68	240.60	151.48	9.61 × 10^6^	264.16	234.12	153.98	9.52 × 10^6^	99.09%
b	307.86	306.12	230.46	2.17 × 10^7^	318.46	305.63	186.58	1.82 × 10^7^	83.61%
c	316.68	245.24	155.22	1.21 × 10^7^	319.66	252.43	155.78	1.26 × 10^7^	104.28%

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
