# Peer review of "A Method for Broccoli Seedling Recognition in Natural Environment Based on Binocular Stereo Vision and Gaussian Mixture Model"

_sensors, 2019, doi:10.3390/s19051132_

Round 1

Reviewer 1 Report

The main remarks of my first review were not properly addressed. The authors made some improvements but not to key problems:

Lack of originality. I agree with the authors that the work has technical/application level originality. From this point of view, depending on the journal policy, it might be acceptable, but:

The matlab implementations (not stated otherwise) of stereo matching are compared (local vs semi-global), and this does not bring anything new for readers that are familiar with fundamentals of stereovision.

The theoretical description of the GMM method on almost 2 pages is useless. As the authors state in the review response, they are just calling a function from the matlab library that does all the work.

Author Response

Response to Reviewer 1 Comments

Point 1: The matlab implementations (not stated otherwise) of stereo matching are compared (local vs semi-global), and this does not bring anything new for readers that are familiar with fundamentals of stereovision.

Response 1: Thank you very much for your correction from the bottom of my heart. I respect your opinion very much. The descriptions of the window size and maximum disparity selection rules, and the influence of different parameter selection on the operation time for SAD and SSD algorithms were all deleted.

From the point of view of engineering application, real-time performance is one of the primary considerations, so GraphCuts algorithm, Dynamic Programming algorithm, and Belief Propagation algorithm were excluded. So the local and semi-global matching algorithms left. SAD, SSD, and SGM have the same parameter selection method, that is to say window size and maximum disparity are need. Which algorithm is the best? Therefore, the stereo matching effect, real-time performance and stability were used to illustrate the advantages and disadvantages of the three algorithms, which were fully demonstrated from the qualitative and quantitative perspectives in this paper. So SGM was selected.

Point 2: The theoretical description of the GMM method on almost 2 pages is useless. As the authors state in the review response, they are just calling a function from the matlab library that does all the work.

Response 2: Thank you again for your correction. I believe that your enlightenment will be of great benefit to my future thesis writing. Following your pertinent opinion, the cumbersome theoretical derivation process has been removed. Instead, the GMM algorithm is introduced in terms of language and formula. Check it please. I am looking forward to your criticism and correction.

Reviewer 2 Report

The authors have carefully answered to all of my comments, generally improving the overall quality of the paper. 

Just two comments. 

1) The first deals with to the following point: "The performance of the 3D reconstruction method is clearly proved, quantified and validated in terms of time and number of points, however the “ground truth” validation is missing.

In other words, the 3D models should be used in order to extract LAI (leaf area index) or volume or number of valid points on the leaves and correlate it with dry weight or actual total leaves area.

Response: As are shown in Figure 2. and Table 2. To further illustrate the completeness of broccoli seeding recognition for the proposed method. The average value of (Intersection area)/(Manual area) is 85.91%. So the proposed method has a good completeness of broccoli seeding recognition."

The authors have compared the projection manually segmented with the 3D: however such approach is somehow acceptable but not scientifically sound. If you propose a 3D method, it has to be validated on a total area (as LAI is) not on a projected area; otehrwise it should be validated against a 3D information (as d.m. mass is). 

2) The second point deals with calibration: "

Calibration has been done on 2D target. But did you do a calibration also on final 3D reconstructed images, against (e.g.) a 3D/freeform calibrated reference standard?

Response: I am sorry for that I didn’t do a calibration also on final 3D reconstructed images, against (e.g.) a 3D/freeform calibrated reference standard. Because 2D target calibration can meet engineering needs, and 3D information of crop can be acquired after 3D reconstruction. I wonder what is “calibration also on final 3D reconstructed images, against (e.g.) a 3D/freeform calibrated reference standard”. Could you tell me please?

A very tipical approach is called "substitution method", as discussed in 

On-barn pig weight estimation based on body measurements by a Kinect v1 depth camera by Pezzuolo et al., Computers and Electronics in Agriculture 148 (2018) 29–36

or 

Traceable volume measurements using coordinate measuring systems. by Carmignato et al. CIRP Ann.-Manuf. Technol. 60 (1), 519–522

Author Response

Response to Reviewer 1 Comments

Point 1: The authors have compared the projection manually segmented with the 3D: however such approach is somehow acceptable but not scientifically sound. If you propose a 3D method, it has to be validated on a total area (as LAI is) not on a projected area; otehrwise it should be validated against a 3D information (as d.m. mass is).

Response 1: Thank you for your pertinent correction. your good academic accomplishment, and your enlightenment are of great benefit to my scientific research and thesis writing. The approach proposed in this paper may not scientifically sound, but it has practical application value.

1)      The 3D point cloud reconstructed by binocular stereo vision is not global point cloud of the whole plane. Because of the mutual occlusion between plant leaves, binocular stereo vision can only reconstruct the visible part of the 2D image at most. It is difficult to determine which part of the plant leaves has been reconstructed, in the actual measurement of the plant blade area which would bring some errors. In addition, the measurement of irregular shape blade area will bring new measurement errors. So an indirect approach was used in this paper.

2)      Considering that the application of machine vision on the intelligent weeding robot and the target spraying equipment is crop identification and setting a protected area or a spraying area around the crop. So the upper projection surface of the crop can be identified completely, that does the work.

Point 2: A very tipical approach is called "substitution method", as discussed in On-barn pig weight estimation based on body measurements by a Kinect v1depth camera by Pezzuolo et al., Computers and Electronics in Agriculture148 (2018) 29–36 or Traceable volume measurements using coordinate measuring systems. By Carmignato et al. CIRP Ann.-Manuf. Technol. 60 (1), 519–522

Response 2: Thank you for dispelling my confused. I have read the papers carefully, and calibration also on final 3D reconstructed images was not done, because actual size (dimension: mm) of crop was obtained through the binocular stereo vision, that does the work.

Round 2

Reviewer 1 Report

The authors have addressed my remarks, although, in my opinion, the GMM description can be simplified even more / or you can replace it with an analysis about how it is applied on your data model (so a particular description - this would be more beneficial for the reader).

Author Response

Response to Reviewer 1 Comments

Point 1: The authors have addressed my remarks, although, in my opinion, the GMM description can be simplified even more / or you can replace it with an analysis about how it is applied on your data model (so a particular description - this would be more beneficial for the reader). 

Response 1: Thank you very much from the bottom of my heart for your pertinent corrections. Improvements of the GMM description have been made in my paper, reflected in the following two aspects:

1)      The derivation details of the formulas are further removed. Instead, the necessary formulas are given directly.

2)      Simple theoretical description of GMM is replaced. Instead, how can the GMM algorithm exactly work on the broccoli seedling point cloud  is illustrated in the paper.

This manuscript is a resubmission of an earlier submission. The following is a list of the peer review reports and author responses from that submission.

Round 1

Reviewer 1 Report

The authors propose a system for detection and recognition of young broccoli plants. The system is based on stereo-matching (SGM), 3D reconstruction, filtering and classification with Gaussian mixtures.

There are two main issues with the paper:

Scientific novelty and content - lacks novelty, and the concepts are poorly presented.

Language and explanations make the paper difficult to follow.

Scientific novelty / content:

The paper might present a level of technical novelty, by the system proposed, but it is based on standard image processing / pattern recognition techniques: calibration (matlab), stereo SGM, etc.

In the introduction there is a classification of stereovision "stereovision can be divided into binocular stereo vision[1,2], 50 multi vision[3,4], RGB-D camera[5,6], structured light[7], multispectral 3D vision system[8,9], laser 51 scanning[10,11] etc". This is quite incorrect, as most of the depth sensors mentioned are not stereovision based. It s a confusion: use "depth sensing can be divided into..."

All the details related to stereovision are basic approaches, no details should be given.

Comparing SGM, a semi-global matching technique, with SAD and SSD local matching techniques is pointless, on several pages of the paper. You do not need numerical evaluations to know that SGM is better, after all that's the reason it was introduced: to provide a denser and more accurate disparity map in contrast with local stereo approaches. 

Page 10: what are Inf points?  "The points of Inf were removed from original"

The Gaussian mixture model is presented from a theoretical point of view, without a clear connection to your data. How do you compute its parameters? How is the recognition performed?

Language and explanations - there are errors (grammar/syntax/formatting), some shown below:

page 2: "because it’s advantages of non-destructive" - its

page 3: "were used to obtaining 3D point cloud" - obtain

"3.3. 3D .reconstruction, invalid points removal and down-sampling results analysis" remove the . before reconstruction

page 12: "Therefor the KNN algorithm" - Therefore

Reviewer 2 Report

See attached comments
